# Immunohistochemical HER2 Recognition and Analysis of Breast Cancer Based on Deep Learning

**DOI:** 10.3390/diagnostics13020263

**Published:** 2023-01-10

**Authors:** Yuxuan Che, Fei Ren, Xueyuan Zhang, Li Cui, Huanwen Wu, Ze Zhao

**Affiliations:** 1Institute of Computing Technology, Chinese Academy of Sciences, Beijing 100190, China; 2School of Computer Science and Technology, University of Chinese Academy of Sciences, Beijing 101408, China; 3Jinfeng Laboratory, Chongqing 401329, China; 4Beijing Zhijian Life Technology Co., Ltd., Beijing 100036, China; 5Department of Pathology, Peking Union Medical College Hospital, Chinese Academy of Medical Sciences and Peking Union Medical College, Beijing 100730, China

**Keywords:** breast cancer, HER2, IHC, whole-slide image, deep learning

## Abstract

Breast cancer is one of the common malignant tumors in women. It seriously endangers women’s life and health. The human epidermal growth factor receptor 2 (HER2) protein is responsible for the division and growth of healthy breast cells. The overexpression of the HER2 protein is generally evaluated by immunohistochemistry (IHC). The IHC evaluation criteria mainly includes three indexes: staining intensity, circumferential membrane staining pattern, and proportion of positive cells. Manually scoring HER2 IHC images is an error-prone, variable, and time-consuming work. To solve these problems, this study proposes an automated predictive method for scoring whole-slide images (WSI) of HER2 slides based on a deep learning network. A total of 95 HER2 pathological slides from September 2021 to December 2021 were included. The average patch level precision and f1 score were 95.77% and 83.09%, respectively. The overall accuracy of automated scoring for slide-level classification was 97.9%. The proposed method showed excellent specificity for all IHC 0 and 3+ slides and most 1+ and 2+ slides. The evaluation effect of the integrated method is better than the effect of using the staining result only.

## 1. Introduction

Breast cancer has become one of the most common cancers worldwide. According to Global Cancer Statistics 2020, there are about 2.3 million new breast cancers worldwide and about 685,000 deaths, accounting for 15.5% of female malignancies [1]. Breast cancer is also one of the important causes of female tumor-related death, which greatly affects the physical and mental health of people all over the world.

Human epidermal growth factor receptor 2 (HER2)-positive breast cancer refers to the amplification of the ERBB2/neu proto-oncogene or the overexpression of the HER2 transmembrane receptor protein. Compared with other types of breast cancer, HER2-positive breast cancer has a high degree of malignancy. It is a special breast cancer subtype with strong aggressiveness, early recurrence and metastasis, and poor prognosis [2,3,4].

HER2 receptor protein overexpression is generally assessed by immunohistochemistry (IHC). Normally, amplification levels of the HER2 gene were detected by fluorescence in situ hybridization (FISH) and chromogenic in situ hybridization (CISH). Guidelines of the Chinese Society of Clinical Oncology (CSCO) 2021 specify the criteria for HER2 to improve the procedures for Her2 testing and standardize the interpretation of the results [5]. According to CSCO 2021, the status of HER2 should be screened by the IHC method first for newly diagnosed breast cancer cases. If the results of the HER2 IHC staining are uncertain, FISH detection should be performed for confirmation. As shown in Table 1, if more than 10% of the infiltrating cancer cells have strong and intact cell membranes with brown staining in IHC slides, the case displays 3+ and it is accepted as HER2-positive. If less than 10% of the infiltrating cancer cells have intact brown cell membranes or more than 10% of the infiltrating cancer cells have incomplete and/or weak to moderate membranous staining, the case is diagnosed as 2+ (HER2-equivocal) and the further ISH testing is needed to assess HER2 expression [6]. If faint/barely perceptible membrane staining is detected in more than 10% of invasive tumor cells, the case is reported as 1+ (HER2-negative). If no staining or faint/barely perceptible membrane staining is seen in less than 10% of invasive tumor cells, the case is reported as 0 (HER2-negative). The IHC evaluation criteria can be summarized into three aspects: staining intensity, circumferential membrane staining pattern, and proportion of positive cells. However, these criteria are still subjective in practice, and there is no specific and explainable numerical basis. Therefore, it is highly significant to propose an interpretable algorithm with automated IHC scoring diagnosis [7].

Computer-aided diagnosis systems have developed rapidly in the medical field [8,9,10,11]. Using computers to perform objective and quantitative analysis of medical imaging data to assist doctors in clinical diagnosis of lesions can help to improve diagnosis accuracy and efficiency [12]. The appearance of digital whole-slide images (WSIs) gives the opportunity to see and analyze more detailed information and make a great step forward in automatic metastatic breast cancer detection [13,14,15,16,17]. WSI is obtained by scanning and collecting traditional glass pathological sections through an automatic microscope or optical magnification system with a digital section acquisition device. It has high resolution and a large file size. In general, WSI has multiple layers, representing a pyramid structure. The different layers of the WSI correspond to different resolutions. The bottom layer of the pyramid has the highest resolution image data, while the upper layers are thumbnails of the bottom image for the pathologist to retrieve the data at low resolutions. It is worth mentioning that the length or width between layers is usually double, which makes downsampling faster and more accurate. However, since a single WSI has billions of pixels, the WSI labeling process is time-consuming for doctors. Therefore, a deep-learning-based network is used for auxiliary analysis of IHC WSIs [18]. Since the computer cannot directly process the WSI image, we need to cut the image into several patches, calculate each patch, then generate a thermal map diagnosis. We proposed an architecture to identify cancer areas in IHC images and generate corresponding probability maps.

## 2. Materials and Methods

As shown in Figure 1, the proposed method includes 3 stages. First, the labeled masks were extracted from the original WSIs with corresponding labels. Then, the tumor patches and normal patches were generated randomly according to label masks. These patches were passed to a deep learning model (ResNet34) to refine the binary classification [19]. In stage two, the tissue mask was extracted from the test WSI. The patches generated from the tissue mask were passed into the model to build the probability map. Then, the binary tumor prediction was produced from the probability map with a threshold. In stage three, the test WSI was differentiated from the four subclasses: IHC 0/1+/2+/3+. This part of the work was implemented to perform an accurate and interpretable result using comprehensive judgments.

### 2.1. Data Acquisition

Pathological slides from breast cancer patients in Peking Union Medical College Hospital from September to December in 2021 were retrospectively included to form the dataset of this study. These slides were scanned into WSIs using Aperio AT2 (Leica, Germany) high-throughput biopsy scanner with a 20-magnification-scale objective and bright-field illumination. The scan resolution was 0.5036 um per pixel. The dataset consisted of a total of 95 whole slide images in Aperio format (.svs). Four categories were included, which were IHC 0, 1+, 2+, and 3+. Two pathologists selected 23 WSIs from the dataset and labeled the area with concentrated and evenly distributed tumor cells for training and testing the deep learning model. The IHC scoring of all pathological images was determined according to CSCO 2021. Figure 2 shows the samples of each category of this dataset. The data distribution performed is shown in Table 2.

### 2.2. Image Preprocessing

Since WSIs usually have billions of pixels, either labeling or calculating WSIs at high magnification scale is an extremely tedious and time-consuming process. By observing the sections, it is found that there are tissue areas and white background areas in each WSI. Therefore, in order to reduce the computation time and complexity, it is necessary to focus on the analysis of the tissue area and skip the white background area. In this paper, a threshold-based segmentation method is utilized to detect the background region automatically. Specifically, the original image is first transferred from the RGB color space to the HSV color space. Then, the Otsu algorithm is used to calculate the optimal threshold of each channel, and the final mask image is generated by combining the masks of the H and S channels [20]. According to the actual calculation, the Otsu method can filter about 75% of the area which belongs to the background, on average, greatly improving the efficiency of calculation.

Due to the limited number of annotated WSI, the image augmentation method can be used to amplify the existing images to increase the type and number of images. Meanwhile, it plays a certain role in inhibiting the overfitting of the model. In this study, images were rotated and flipped at random angles so as to increase the number of training patches. It is worth noting that, in the process of image extraction or image augmentation, staining standardization is not carried out. The reason is the existing dyeing standardization algorithm with good effect is not perfect in the call of GPU, and it takes too long to process a single-patch image (1–7 s/patch). In addition, staining standardization changes the color of the image to some extent. Without a color standardization process, the brown part and the blue part of IHC staining can still be distinguished by color space conversion.

Before the algorithm training was initiated, the 20-magnification-scale WSIs were cropped into 256 ∗ 256 pixels of patches, and 23 labeled WSIs were randomly divided into the training set and the test set. A total of 8000 patches were obtained from the training set. Lesions identified on these patches were utilized to train and test the performance of ResNet34 algorithm. Annotated patches of training and validation datasets were separated by the function embedded in the scikit-learn package (ratio 9:1), and total of 16,000 images were eventually used in stage one, among which 14,400 images were used for training and 1600 images were used for validation.

### 2.3. Deep Learning Structure

Deep learning has been extensively used in the diagnosis and analysis of medical images in recent years [21,22,23,24,25,26,27]. The convolutional neural network stands out among many deep learning networks because of its strong feature learning ability and has become a cutting-edge algorithm in the field of image classification. In stage one, ResNet34 is used as the deep learning backbone network in this paper. During the training process, 256 ∗ 256 ∗ 3 patches from the tumor and non-tumor regions of WSIs were used as inputs in this stage to train the classification model to distinguish the two classes. During the training phase, the hyperparameters of the network were set as follows: the optimizer was set to SGD, the learning rate was set to 0.01, the momentum was set to 0.9, the loss function was set to nn.CrossEntropyLoss, the epoch was 50, and the patch size was 64.

### 2.4. Extraction of Membranes and Cells

The diagnosis of HER2 dominates the type of subsequent treatment. Therefore, this diagnosis becomes very important for breast cancer patients [28]. IHC is a special staining method for finding the HER2 protein in cancer cells based on the detection of specific antigens in tissue. The IHC staining slides are composed of a brown channel (diaminobenzidine, DAB signal) and a counterstain blue/violet channel (hematoxylin, H signal). The membrane extraction method used in this study was performed on the brown channel.

According to the CSCO 2021 guidelines, the evaluation criteria of IHC-stained slides is directly related to the membrane staining condition. Therefore, the staining intensity is presented as an evaluation indicator. Staining intensity is the most important feature for the classification of HER2 slides. The staining intensity indicates the depth of a certain color in the image. In this study, it refers to the depth of brown areas. The image is converted from RGB color space to HSV color space, and the brown area is extracted by a function in OpenCV library [29]. After that, the extracted brown areas are converted to gray level and its depth is calculated. When the staining intensity is low, it indicates that the extracted area is lightly stained, while when the staining intensity is high, it indicates that the extracted area is deeply stained. Its value range from 0 to 1. The staining intensity of each IHC score is in a different range, and hence this evaluation index is a good candidate feature for further classification [30,31,32,33].

In this study, color deconvolution and watershed algorithms are employed to extract tumor cells. After the input RGB image is preprocessed, the RGB value is converted into optical density (OD) space, with the value range of [0, 1]. The inverse matrix of the OD matrix is the required deconvolution matrix [34]. Therefore, color deconvolution method is employed to separate and distinguish DAB and H staining [35]. Due to the overlap between cells, the watershed algorithm based on distance transformation is needed in order to process the binary image obtained by the color deconvolution method. The specific algorithm is as follows.

Firstly, the image after color deconvolution is converted to gray level and the morphology operation is performed to eliminate the interference on the boundary. Then the distance transformation of the gray image is carried out to split the adherent cells. Finally, expansion and filling methods are performed, and the connected component-based method is used to extract the cells.

After the results of staining and cell extraction are obtained, the three interpretative evaluation indicators of IHC scoring are calculated using these results. The flow diagram of the specific algorithm is shown in Figure 4. The result of staining intensity can be calculated from the mean value of the results extracted by staining. In addition, a proper threshold should be set for the results of staining extraction for morphological corrosion and expansion operation. The number of positive cells can be obtained by contour extraction. The circumferential membrane staining pattern can be obtained by dividing the number of positive cells calculated above by the total number of cell counts. For the calculation of the proportion of positive cells, traversing the pixels of the stained area to find the nearest cell of these pixels for calculation is an accurate method. However, the algorithm has high computational complexity and greatly increases the running time. In view of this, this study presents a rapid method to calculate the proportion of positive cells. The core of this method is to multiply the stain extraction mask and the cell extraction mask. If the result of one pixel is non-zero, the cell where the point is located is marked as a staining-positive cell. Otherwise, it is marked as a staining-negative cell. This algorithm has low computational complexity. In addition, the computational efficiency greatly improves.

## 3. Results

PyTorch was used to build and form the CNN model in this study. All experiments were conducted with a Linux server (Linux version CentOS 3.10.0 to 69.3.EL7.x86_64, CPU version Intel (R) Xeon (R)Silver 4114 @ 2.20 GHz. The name of the graphics card is NVIDIA GeForce RTX2080 Ti).

### 3.1. Tumor Area Classification

Figure 5 visualizes the training results of the stage one. Figure 5a is the original WSI. The black outline in Figure 5b shows the tumor areas marked by the pathologist. Due to the high resolution of WSI, patch-level classification of images is performed. The probability heat map of the tumor region output by the deep learning model is shown in Figure 5c with the original image added. Dark red color indicates areas with high probability of cancer, while light and blue areas indicate low probability of tumor. The results are basically in line with the expert annotations. A binary image of tumor areas is produced by selecting a proper threshold of the probability map. Note that small areas are filtered for visualization during the tissue area extraction phase.

The dataset with annotations is randomly divided into the training set and the test set. At stage one, 16 WSIs are used for model training, and the remaining labeled WSIs are used for model inspection and evaluation. The following evaluation indexes are used to evaluate the model performance [36,37,38,39,40,41].
(1)$$Accuracy=\frac{TP+TN}{TP+FP+TN+FN}$$
(2)$$Precision=\frac{TP}{TP+FP}$$
(3)$$Recall=\frac{TP}{TP+FN}$$
(4)$$F1score=2\times \frac{Precison\times Recall}{Precision+Recall}$$
where *TP* is true positive cases, *FP* stands for false positive cases, *TN* is true negative cases, and *FN* represents false negative cases. Table 3 shows the average results of patch-level data analysis. The values of false positive rate and true positive rate of the receiver operating characteristic (ROC) curve is shown in Figure 6a. The closer the ROC curve to the upper left-hand corner and the larger the area under curve (AUC) value, the better the performance of classification. Similarly, the closer the precision-recall (PR) curve to the upper right corner and the larger the average precision (AP) value, the better the performance of the model. It can be calculated that the proposed model for tumor classification provides high value of AUC (0.983) and AP (0.984).

### 3.2. IHC Slide Classification

In stage three, the tumor areas in each WSI need to be further processed. Firstly, downsampling is performed on tumor areas. The rate of downsampling needs to be calculated to extract non-overlapping patches of 512 ∗ 512 pixels. For each patch, the staining area and the number of cells is extracted. Figure 7 shows the extraction of staining regions of different IHC score. It can be observed that with the increase of IHC score, the color extracted from the staining area is darker. Therefore, the initial classification of pathological slides is constructed based on staining area by setting the proper threshold with this feature. It is also the main reference basis for IHC scoring.

In this study, color deconvolution and watershed algorithm are adopted for cell extraction. For the cohesive and overlapping cells, the method based on the single-cell area is used for segmentation and statistics. As for the extraction of cells with circumferential positive membrane, a patch with IHC 3+ score is considered as an example. The schematic diagram is shown in Figure 8. The confusion matrices of slide-level IHC results which only used staining as the scoring basis and integrated the three scoring methods are illustrated in Figure 9a (accuracy: 87.4%) and Figure 9b (accuracy: 97.9%), respectively. In addition, we calculate the average time consumption of staining only method and integrated method to compute a single patch. The comparison of time cost between these two methods is shown in Figure 10. In general, the time cost of integrated method is nearly twice as much as staining only method. In addition, Table 4 shows the statistical calculation of evaluation indexes. For a single center dataset, the experimental thresholds can be set via this table.

## 4. Discussion

This study proposed a deep-learning-based predictive framework to automatically evaluate the IHC score of HER2 WSIs in breast cancer. In total, 95 IHC section images with 23 labeled tumor areas were provided by Peking Union Medical College Hospital. The corresponding IHC scores of all HER2 slides were used in the study of this project. The predictive framework of this study is more interpretable compared with previous HER2 evaluation methods.

When a WSI is tested, the tumor area of this image is first predicted. Then, the evaluation index in tumor area is calculated and analyzed. As can be seen from the results in Table 1, the patch-level precision and f1 score of the proposed model are 95.77% and 83.09%, respectively, showing good performance in similar studies.

In stage three, three evaluation indexes of tumor patches were calculated. These calculated features are identified based on CSCO 2021 guidelines, which are referenced by pathologists when scoring IHC manually. The proposed method in stage three can extract stained areas, extract positive cells, and extract cells with complete positive cell membranes. It achieves high slide-level accuracy of HER2 score according to evaluation indexes. It is worth mentioning that the selection of the threshold is intuitive and important. Certain knowledge is needed to adjust a better threshold. Improper thresholds seriously affect the accuracy of the method. Figure 8 presents the results of cell extraction and positive cell labeling. These results are roughly consistent with the determination by manual observation, indicating the high feasibility of the prediction framework. In Table 5, we compare our method with other relevant methods. Most of these methods use manually labeled WSIs to train and test models. The proposed method can perform HER2 scoring based on WSI and obtain a high scoring accuracy.

There are some limitations to our study. First, there may be inconsistencies in the depth of the color due to inconsistencies in the dose of dye added during the slicing process. Some deep learning methods use color standardization to better identify tumor areas. Generally, color normalization was added in the image preprocessing stage of model training and testing to make the result of classification more accurate. However, after the addition of this method, the time of execution is greatly increased, and the higher accuracy has a limited impact on the accuracy of the final IHC score. Secondly, although the processing method in stage three has a high dependence on the selection of color thresholds, this paper finds that the overall robustness of color extraction is stable after the conversion of color space. In addition, the influence of color depth on classification results can be limited by collecting HER2 slides from multiple centers.

In conclusion, this study conducted interpretable analysis and prediction of IHC scores of histological images of HER2 slides based on the deep learning method. It provides the direction for the clinical application of deep learning and promotes the development of precision therapy in the field of breast cancer.

## Figures and Tables

**Figure 1 diagnostics-13-00263-f001:**
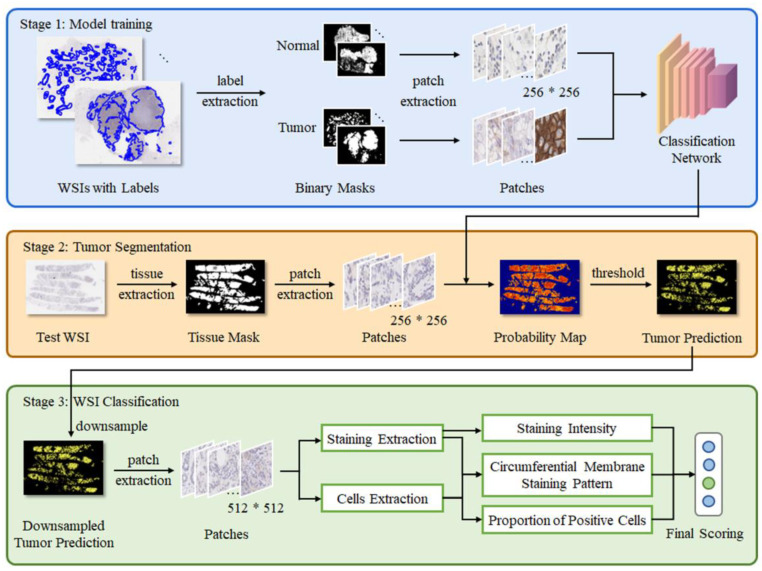
The proposed framework for this study.

**Figure 2 diagnostics-13-00263-f002:**
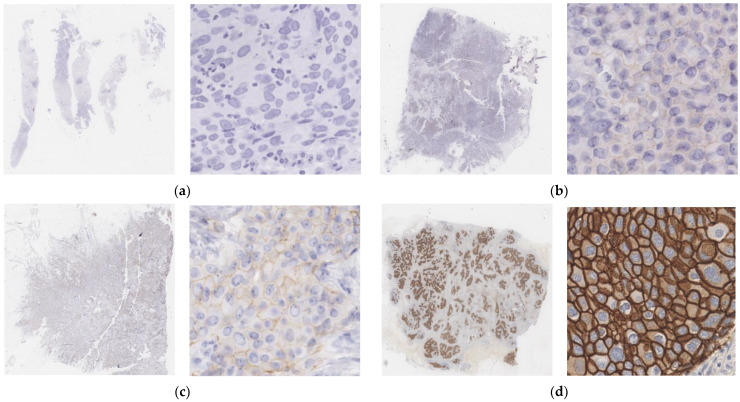
Sample thumbnail images (**left**) and patches (**right**) of the dataset for four categories showing typical levels of membrane staining. (**a**) IHC 0. (**b**) IHC 1+. (**c**) IHC 2+. (**d**) IHC 3+.

**Figure 3 diagnostics-13-00263-f003:**
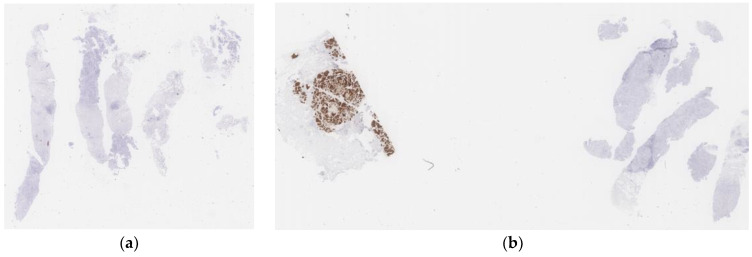
(**a**) Normal WSI. (**b**) WSI with 3+ control.

**Figure 4 diagnostics-13-00263-f004:**
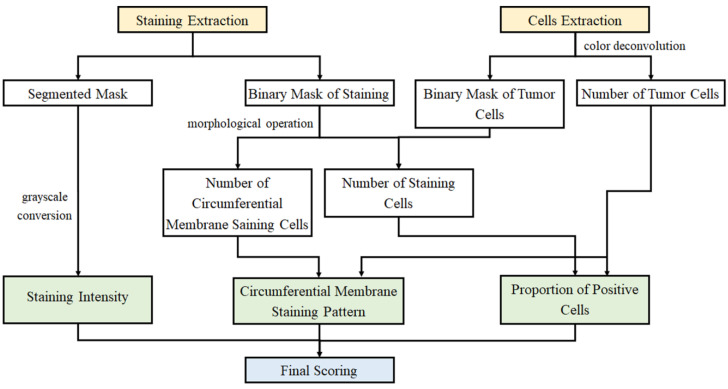
The detailed flow diagram of stage three.

**Figure 5 diagnostics-13-00263-f005:**
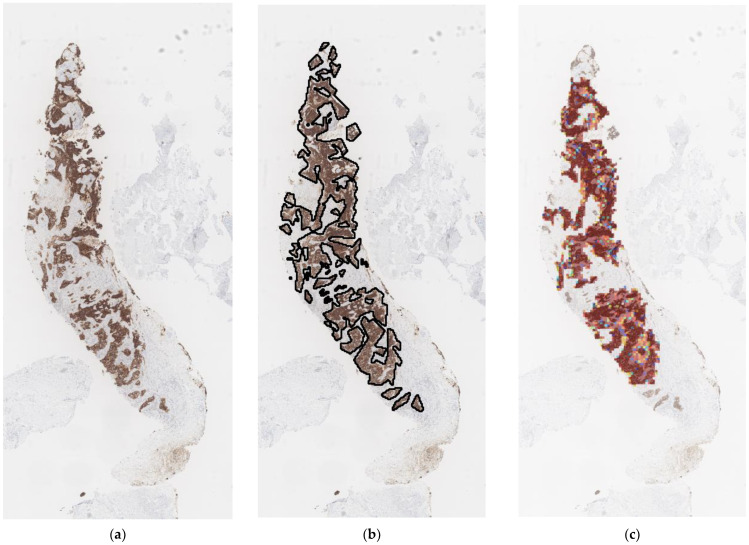
Visualization of model training result. (**a**) Original WSI. (**b**) WSI with black annotation. (**c**) The patch-level classification of tumor areas.

**Figure 6 diagnostics-13-00263-f006:**
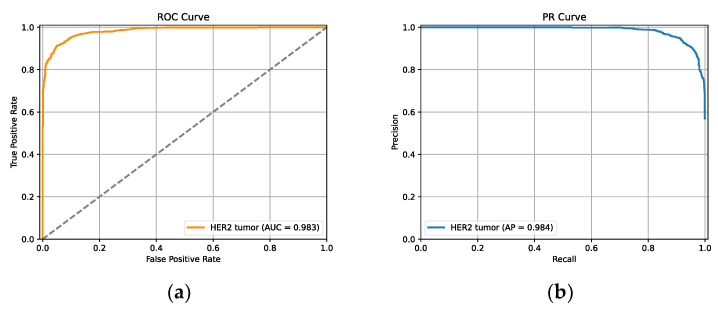
The ROC curve and PR curve give an estimation of the cell segmentation model’s performance. (**a**) ROC curve. (**b**) PR curve.

**Figure 7 diagnostics-13-00263-f007:**
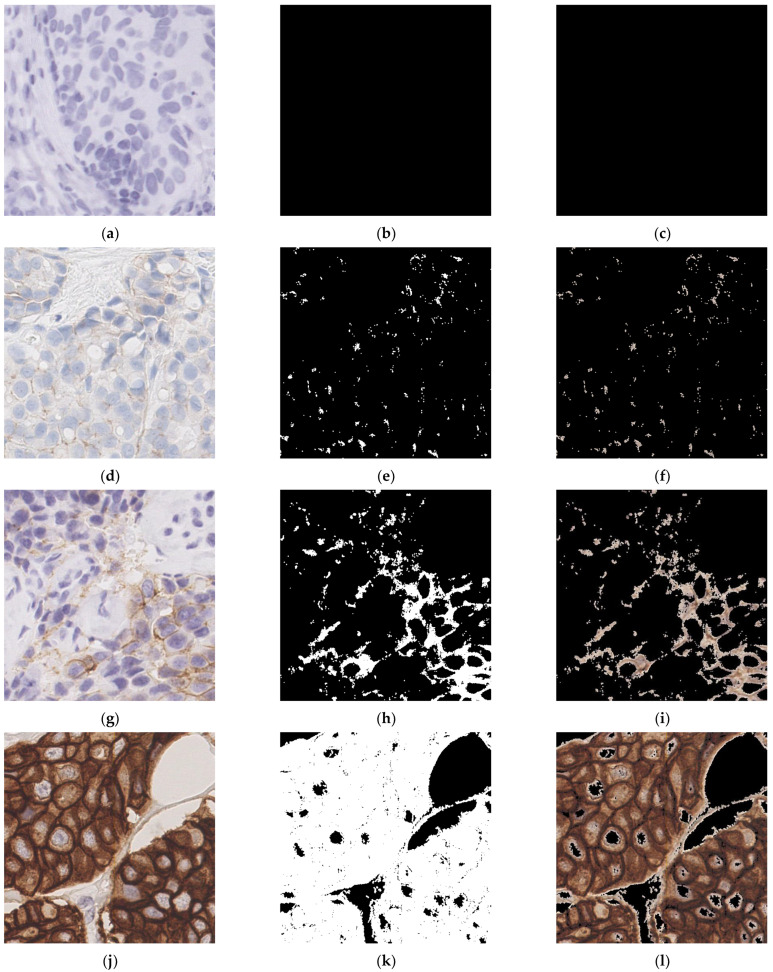
Visualization of staining region extraction. Each row includes the original patch (**left**), the binary image of extracted staining (**middle**), and the image of staining area with original image superimposed (**right**) for one IHC scoring. (**a**–**c**) IHC 0. (**d**–**f**) IHC 1+. (**g**–**i**) IHC 2+. (**j**–**l**) IHC 3+.

**Figure 8 diagnostics-13-00263-f008:**
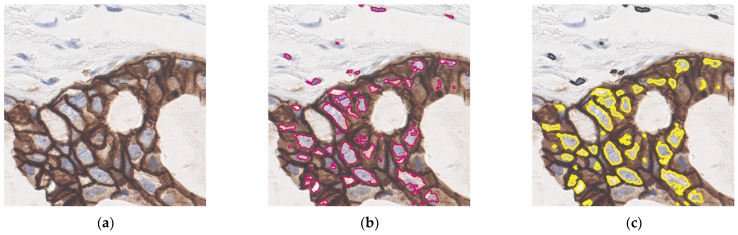
Visualization of cell extraction result. (**a**) Original patch. (**b**) Image with all extracted cells (red contours). (**c**) Divide the extracted cells to positive cells (yellow contours) and negative cells (dark blue contours).

**Figure 9 diagnostics-13-00263-f009:**
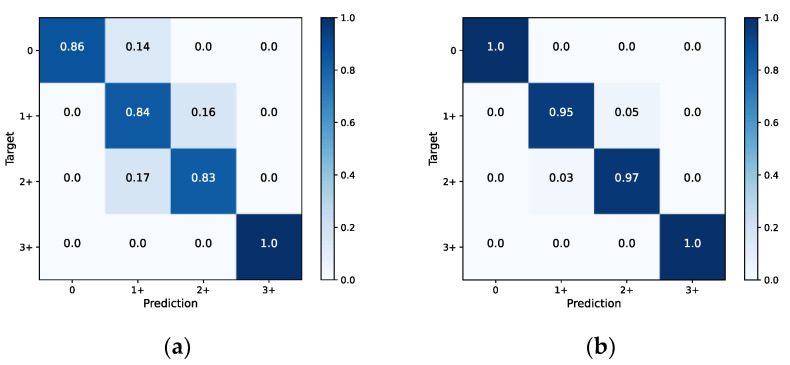
Confusion matrices of HER2 slide-level classification. (**a**) Using staining as the classification method only. (**b**) Using integrated methods.

**Figure 10 diagnostics-13-00263-f010:**
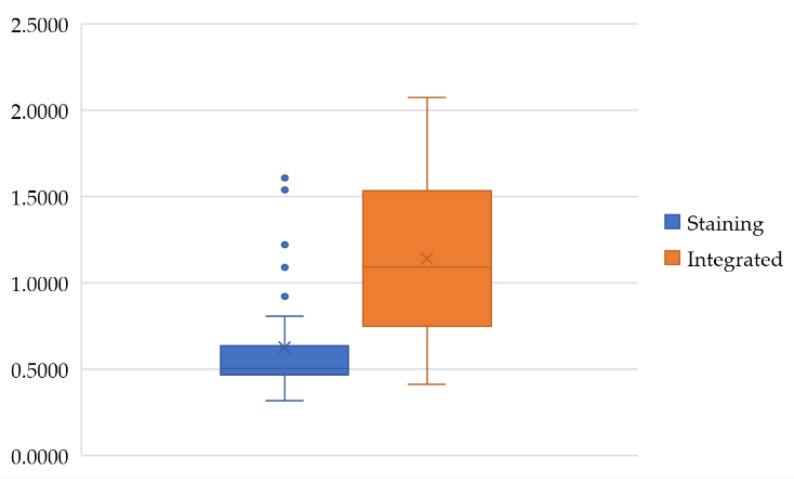
The time/cost comparison between staining method and integrated method for automated scoring.

**Table 1 diagnostics-13-00263-t001:** Evaluation criteria for HER2 expression by IHC assay in breast cancer.

IHC Score	Staining Pattern	HER2 Expression
0	No staining or incomplete membrane staining which is faint or barely perceptible in ≤10% of invasive tumor cells	Negative
1+	Incomplete membrane staining which is faint or barely perceptible in >10% of invasive tumor cells	Low expression
2+	(a) Weak to moderate membrane staining with uneven brownish yellow coloration in >10% of invasive tumor cells (b) ≤10% of invasive tumor cells have circumferential membrane staining which is complete, intense, and has brownish coloration	Equivocal (low expression if the slide is ISH-negative, positive if it is ISH-positive.)
3+	>10% of invasive tumor cells have circumferential membrane staining which is complete, intense, and has brownish coloration	Positive

**Table 2 diagnostics-13-00263-t002:** Composition of the dataset.

IHC Score	No. WSIs	No. Labeled WSIs ^1^	No. WSIs with 3+ control ^2^
0	14	2	6
1+	25	7	7
2+	36	7	24
3+	20	7	10
Total	95	23	47

^1^ The labeled areas are all tumor areas rather than regions which only satisfy the IHC score. ^2^ WSI with 3+ control means this WSI has a IHC 3+ control tissue next to the main tissue (see Figure 3).

**Table 3 diagnostics-13-00263-t003:** Patch-level classification performance on the test set.

Evaluation Indexes	Accuracy	Precision	Recall	F1 Score
	73.49%	95.77%	73.38%	83.09%

**Table 4 diagnostics-13-00263-t004:** The statistical calculation of evaluation indexes.

IHC Score	0	1+	2+	3+
Staining average min	0.000	0.006	0.146	0.541
Staining average max	0.008	0.189	0.367	0.611
Positive cell ratio min	0.000	0.004	0.130	0.173
Positive cell ratio max	0.041	0.125	0.318	0.338
Circumferential membrane cell ratio min	0.000	0.011	0.454	0.461
Circumferential membrane cell ratio max	0.000	0.360	0.741	0.908

**Table 5 diagnostics-13-00263-t005:** The comparison with related works.

	Dataset	Method	Remarks
Saha et al. [24]	752 labeled images cropped from 79 WSIs	Fully connected long short-term memory network, scoring by membrane and nuclei detection	98.33% accuracy
Vandenberghe et al. [27]	74 WSIs	Watershed segmentation, support vector machine, random forest, scoring by classifying cells	83% accordance
Qaiser et al. [26]	86 WSIs	Deep reinforcement learning, scoring by connectivity-based method	79.4% accuracy
Singh et al. [38]	1345 labeled areas from 52 WSIs	Neural network classifier, scoring by ROI-based method	91.1% accuracy
Caroline et al. [39]	2580 labeled images from 86 WSIs	K-nearest neighbor, multilayer perceptron, scoring by decision trees	90% accuracy
Khameneh et al. [41]	127 WSIs	Modified U-Net, scoring by WSI merging and membrane segmentation	94.82% segmentation and 87% classification accuracy
The proposed method	95 WSIs	ResNet, WSI segmentation, scoring by integrated calculation of staining intensity, circumferential membrane staining pattern, and proportion of positive cells	73.49% segmentation accuracy, 95.77% segmentation precision, 97.9% scoring accuracy

## Data Availability

The data are not available for public access because of patient privacy concerns but are available from the corresponding author on reasonable request.

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
