# Peer review of "Immunohistochemical HER2 Recognition and Analysis of Breast Cancer Based on Deep Learning"

_diagnostics, 2023, doi:10.3390/diagnostics13020263_

Round 1

Reviewer 1 Report

your work is interested, and some modifications are needed to be suitable for publication :

1. Refer what is WSI 

2. The number of images is too low to make a trustable deep learning software, so do you have any ability to expand your work by getting involved more images.

3. Increase the number of references 

4. Enhance the visualization of confusion matrices 

5. Add more performance curves such as ROC and PR curve.

6. Compare your results and approach with literature.

7. Did you augment your patches to be compatible with CNN. 

Author Response

Response to Reviewer 1 Comments

Point 1: Refer what is WSI

Response 1: Thank you for your kind advise. We have supplemented WSI in introduction section.

Point 2: The number of images is too low to make a trustable deep learning software, so do you have any ability to expand your work by getting involved more images.

Response 2: We are grateful for your kind question. As we mentioned in our paper, our dataset has 95 WSIs in total. These slices were provided by Peking Union Medical College Hospital. In stage 1, we firstly extracted patches for network training from labeled WSIs. We extracted 16000 patches for each class (tumor and normal tissue), which should be enough to train a deep learning network. We have tried to extract more patches (40000 patches in total, 20000 patches for each class), and we found limit improvement in test results compared with former model. To expand the number of patches, we used multiple measurements to enhance patches such as rotate, flip, etc.

Point 3: Increase the number of references

Response 3: Thank you for your kind recommendation. We have increased the number of references in the revision script.

Point 4: Enhance the visualization of confusion matrices

Response 4: We appreciate for your kind advise. We have enhanced the visualization of confusion matrices to make the colors more distinguishable and clearer.

Point 5: Add more performance curves such as ROC and PR curve.

Response 5: We are grateful for your kind suggestion. We have added ROC curve as well as PR curve of the training model in the revision script.

Point 6: Compare your results and approach with literature.

Response 6: We appreciate for your kind advice. We have added the comparation of results in the revision script (see Table 5).

Point 7: Did you augment your patches to be compatible with CNN.

Response 7: We are grateful for your kind question. All the augmented patches were compatible with CNN. The original size of these patches was 256*256. They were cropped into 244*244 before they were input to CNN.

Reviewer 2 Report

Dear authors,

I have read the manuscript and have only few questions for you:

1) Please can you add a statistical calculation in order to evaluate the number of sample necessary to obtain a significant result

2) please can you add a table showing the difference in time of work and costs between the classification method and the integrated method 

Author Response

Response to Reviewer 2 Comments

Point 1: Please can you add a statistical calculation in order to evaluate the number of sample necessary to obtain a significant result

Response 1: We appreciate for your kind suggestion which is of great practical significance. We have added the statistical calculation of three main indexes in the revision script (see Table 4). In order to set better thresholds, we usually use all the slices to evaluation. At the same time, we think that current manuscript can still support the argument of this article. Therefore, we recommend that the supplementary evaluation be included in another follow-up paper in the future.

Point 2: Please can you add a table showing the difference in time of work and costs between the classification method and the integrated method

Response 2: We are grateful for your kind question. We have added the statistical calculation in the revision script (see Figure 10).

Round 2

Reviewer 2 Report

Dear Authors,

I have read with interest the manuscript and I think that it has been imporved and can be accepted.